# A City Surveillance System for Social Health Inequalities: The Case of Barcelona

**DOI:** 10.3390/ijerph20043536

**Published:** 2023-02-17

**Authors:** Carme Borrell, Laia Palència, Marc Marí-Dell’Olmo, Xavier Bartoll, Mercè Gotsens, M. Isabel Pasarín, Lucía Artazcoz, Maica Rodríguez-Sanz, María José López, Katherine Pérez

**Affiliations:** 1Agència de Salut Pública de Barcelona, 08023 Barcelona, Spain; 2CIBER de Epidemiología y Salud Pública (CIBERESP), 28029 Madrid, Spain; 3Institut de Recerca Biomèdica Sant Pau (IIB SANT PAU), 08041 Barcelona, Spain; 4Departament de Ciències Experimentals i de la Salut, Universitat Pompeu Fabra, 08003 Barcelona, Spain

**Keywords:** urban health, health inequities, public health surveillance

## Abstract

Introduction: In the past, health inequalities were not prioritised in the political agenda of Barcelona. The change of city government (2015) was an opportunity to develop a Surveillance System for Social Health Inequalities in the city, which is described in this article. Methods: The design of the Surveillance System formed part of the Joint Action for Health Equity in Europe (JAHEE), funded by the European Union. Various steps were considered by the experts to set up the System: define its objectives, target population, domains and indicators, and sources of information; perform data analysis; implement and disseminate the system; define the evaluation; and perform regular data updates. Results: The System considers the following domains: social determinants of health, health-related with behaviours, use of healthcare, and health outcomes, and includes eight indicators. As axes of inequality, the experts chose sex, age, social class, country of origin, and geographical area. The Surveillance System for Social Health Inequalities is presented on a website including different types of figures. Conclusion: The methodology used to implement the Surveillance System can be used to implement similar systems in other urban areas around the world.

## 1. Introduction

The social determinants of health are the conditions under which individuals are born, grow, work, live, and age, and these determinants cause inequalities in health in the sense that not all populations have the same opportunities to live with good health and quality of life. Taking action on inequalities in health is recognised as crucial for reducing the growing health inequalities that exist within and between countries. Monitoring these inequalities is an important step to raise awareness of health inequalities and to introduce them into the political agenda. Moreover, a monitoring system would ensure that health inequalities could be tracked across space and time and, therefore, making it useful to promote an equitable society [1,2]. However, recently Högberg et al. [3] stated that few European countries have a proper health inequality monitoring strategy in place and, when they do exist, they often do not coincide with policymakers’ needs.

Urban areas have important social inequalities which result in health inequalities, and it is important to closely follow their evolution across time and space in order to monitor them. In the city of Barcelona (Spain), public health researchers have produced a wealth of research showing health inequalities by gender, social class, geographical area, and other axes of inequality [4,5]. Moreover, a conceptual framework was developed and published to define the social determinants of health in urban areas [6]. Some years ago, the proposal of the World Health Organization, the ‘Urban Health Equity Assessment and Response Tool (URBAN-HEART)’ [4], was adapted for the city and its neighbourhoods, although this tool was not updated.

However, in the past, health inequalities were not prioritised in the political agenda [5]; therefore, a surveillance system was neither available nor planned. In 2015, the change of city government was an opportunity to advance the inclusion of health inequalities in the political agenda. This has been shown by the Government Measure to Reduce Social Health Inequalities in Barcelona [6] and the creation of the Observatory of Health and Impact of Policies [7]. Finally, this allowed us to develop the Surveillance System for Social Health Inequalities in the city of Barcelona, which is described in this article.

## 2. Establishment of the Surveillance System

The design of the Surveillance System formed part of the Joint Action for Health Equity in Europe (JAHEE), a joint action including 25 European countries and led by the Instituto Superiore di Sanita of Italy [3,8]. The objective of JAHEE was to improve the health and well-being of European citizens and achieve greater equity in health outcomes. This Joint Action had many Work Packages (WP), where the objective of WP 5 was to support the countries in developing a monitoring system on health inequalities adapted to national contexts, well suited to policy requirements and sustainable over time. On behalf of the Spanish partners working on WP 5, the research team from the Agència de Salut Pública de Barcelona (Public Health Agency of Barcelona) had the objective of developing the Surveillance System for Social Health Inequalities in the city.

Various steps were considered to set up the System: define its objectives, target population, domains and indicators, and sources of information; perform data analysis; implement and disseminate the system; define the evaluation; and perform regular data updates [2,9].

To define and carry out these steps, a group of experts was established inside the Public Health Agency of Barcelona. This group is formed by professionals specialising in public health with different fields of expertise (epidemiology, statistics, economy, and medicine). All of them have broad experience in analysing and doing research on health inequalities [4,5]. The group of experts met several times and decided by consensus upon the different aspects of the system, considering other European systems, mainly those described by the JAHEE Joint Action [3]. Moreover, the advances made regarding decisions were shared with the researchers of WP 5 of JAHEE.

## 3. The Surveillance System

Barcelona is a Mediterranean city that has some 1,700,000 inhabitants, 10 districts (mean population in 2021: 166,031), and 73 neighbourhoods (mean population in 2021: 23,050).

Objective of the Surveillance System for Social Health Inequalities: The system aims to monitor health inequalities in Barcelona.

Target population: The System included all residents of Barcelona.

Domains and indicators: The following domains are considered: social determinants of health, health-related with behaviours, use of healthcare, and health outcomes. The experts considered that, for the System to be simple and effective, it should have a low number of indicators. Such indicators should be: (a) valid to measure the results; (b) able to reflect health inequalities; (c) possible to be shown by area and other axes of inequality; and (d) periodically obtained from routine data sources. The indicators were chosen to represent the domains selected and were based on previous reports and the exchanges made between the experts of the JAHEE project [3,9].

The Surveillance System includes eight indicators (see Table 1). As a social determinant of health, the experts chose an indicator that lets socioeconomic inequalities be described across the districts and neighbourhoods of the city: the index of each area’s income. The remaining seven indicators are: one health indicator (overweight and obesity) related with behaviours such as diet and physical activity; one for the use of healthcare, trying to show an aspect of health services not included in the National Health System and therefore causing inequalities in accessing it (having visited the dentist); and five for health outcomes (perceived health status, perceived mental health status, life expectancy at birth, teenage pregnancy, and COVID-19 incidence). When possible, the indicators should be stratified by axes of inequality such as sex, age, social class, country of origin, and geographical area. Social class was measured by considering the occupation reported in the health interview surveys. To classify the occupations, the proposal of the Spanish Society of Epidemiology [10] was followed, and people were divided into five classes, from class I (most advantaged) to V (least advantaged).

Sources of information: The sources of information are reported in Table 1: the Barcelona Health Survey (ESB) and the Catalan Health Survey (CHS), both based on interviews with a representative sample of the population of Barcelona. The ESB and CHS surveys are both official surveys approved by the Catalan Institute of Statistics and share a common methodology in survey sampling and in the field and quality processes. For ESB, a random non-proportional sampling of 400 persons per district was carried out in sex and age quotas (10 districts). Non-responses are replaced with population of the same quota until the target sample is reached. For example, for 2021 the total sample was 4000 persons of which 3556 were over 14 years old. This sample design lets the indicators be estimated with a margin of error of +/−1.58% at the aggregate level of Barcelona and +/−5.6% per district. For CHS, a random proportional to district population is sampled by quotas of sex and age (10 districts). For instance, in 2019 a total of 1343 adults were interviewed. Non-responses are replaced with population from the same quota until the target sample is reached. To increase the sample size, we collapsed two consecutive years, for instance 2018 and 2019, to reach a total of 1896 adults, which implies an aggregate error for Barcelona of +/−2.2%.

Different population health registers of Barcelona residents were used: mortality, natality and abortions, based on the certificates of deaths, births and abortions, respectively. In addition, the COVID-19 register based on data collected from all COVID-19 cases of Barcelona residents. Moreover, the income of the district or neighbourhood was obtained from the Barcelona City Council.

Data analysis: Measures are described in Table 1. To analyse inequalities by social class, robust Poisson regression models were adjusted. The relative index of inequality (RII) was calculated. The RII value can be interpreted as the ratio of the health outcome indicator between the two extreme social classes, the least and the most advantaged. Absolute differences of the outcomes between the two extremes of the social class spectrum were estimated calculating the slope index of inequality (SII) [13]. Prevalence ratios or prevalence differences were used as measures of association of inequalities by axes with 2 categories (sex, age, and country of origin). Analyses were performed using R software [14].

Implementation of the Barcelona Social Health Inequalities Surveillance System: Data have been mainly presented through a website [15]. It has been developed with the R Markdown package, which ensures easy updating of the data and the results, and their visualisation. The outcomes obtained from the Health Surveys are presented in figures showing histograms by axes of inequality or measures of association and their 95% confidence interval. For the outcomes obtained from registers (life expectancy, teenage pregnancy, and COVID-19), the information cannot be presented for all axes of inequality. However, as they have a greater number of cases, the outcome can be presented across areas by drawing maps and showing the values by quintiles of income. Depending on the availability of the data, trends are presented since the beginning of this century. The results are also stratified by sex. An example of some of the information produced can be seen in Figure 1.

Dissemination: Apart from the JAHEE regular meetings, the dissemination has mainly been done through the website launched in October 2021 and updated in June 2022. Moreover, the System has been presented at several scientific congresses and lectures at the Public Health Agency of Barcelona (included on the organisation’s YouTube channel).

Evaluation: The System will be evaluated by analysing its utility in detecting the evolution of health inequalities and the adequacy of the indicators each year. In the Public Health Agency of Barcelona there are other health information systems [5] that will reveal if other health inequalities arise, showing the need to update the system. The use of the website will be periodically monitored, together with data presentation in specific reports. Analyses will also be conducted to determine whether the System is useful for following specific health objectives or city interventions. Moreover, different attributes of the system will be analysed, such as simplicity, flexibility, data quality, and possibility to be updated.

Regular data update: Data will be periodically updated, preferably each year.

## 4. Concluding Remarks

The Surveillance System for Social Health Inequalities in the city of Barcelona has been implemented since the end of 2021. We believe that the system will show a general overview of health inequalities in Barcelona across different areas and years, and become of interest to politicians, managers, technicians, city entities, and citizens.

Following the example of the Marmot Indicators for local authorities in the UK (that includes 15 indicators [16], as well as other experiences [3,17], the group of experts chose a small number of indicators. Other information systems of the city contain more [5,7], but in our case simplicity is crucial to ensure that the data can be updated regularly. Moreover, as each indicator is shown for the different axes of inequality (sex, age, country of origin, social class, neighbourhood, and district) every year, there is therefore a lot of information to be presented and the inclusion of more indicators would not favour the usability of the system.

As for the specific indicators, only one social determinant of health was included (income) to show health inequalities by district or neighbourhood, because its distribution across these areas is similar to other socioeconomic indicators. The study of these geographical areas has been useful for prioritising interventions and reducing social health inequalities. As stated above, the indicator of inequalities in the use of healthcare is having visited the dentist because most dental treatments are not covered by the Spanish National Health System. Therefore, having visited a dentist reflects social inequalities. However, it is necessary to take into account that if other health information systems, such as the Health Interview Survey of Barcelona, detect inequalities in access to health services [5], the monitoring system can include new indicators in the future. Finally, the COVID-19 incidence indicator is likely to be removed in the near future depending on the evolution of the disease and it can be substituted by another infectious disease, for example tuberculosis, which also affects the most vulnerable populations. However, we considered it important to include COVID-19 in the Surveillance System, since the pandemic caused significant COVID-19 inequalities that have been varying across the different waves [18,19].

The Public Health Agency of Barcelona has the advantage of having the datasets and the professional skills to set up a Monitoring System as it is presented. This System will be very useful to determine how health inequalities are evolving over time and space and therefore to strengthen or implement policies to tackle these inequalities in the whole city or in its neighbourhoods or districts. However, in the future the system can have limitations, such as not being able to show inequalities or that the indicators are not useful anymore. But it merits mention that the system is flexible and can also include new indicators according to new realities or other axes of inequality (e.g., unemployment). In the future, we could also get real-time data or link different registers.

The Surveillance System for Social Health Inequalities in the city of Barcelona has been developed in the Joint Action for Health Equity in Europe. As many countries have participated in this joint action, the design and the final product have been presented in several international meetings. These aspects have probably influenced both the quality of the system and the interest from other countries and cities. 

## 5. Conclusions

We have presented the Surveillance System for Social Health Inequalities in the city of Barcelona. As monitoring systems in cities have hardly been implemented, the methodology used to implement the Surveillance System in Barcelona can be useful for governments or other stakeholders in order to implement similar systems in other urban areas around the world, although the indicators will probably be different.

## Figures and Tables

**Figure 1 ijerph-20-03536-f001:**
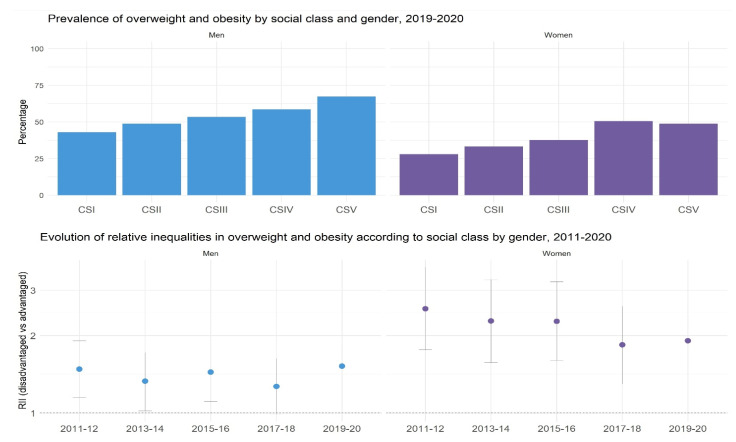
Example of figures included in the Barcelona Social Health Inequalities Surveillance System. CS: social class, where I is the most advantaged, V the least advantaged, RII: Relative index of inequality comparing social classes. The figure includes population ≥ 15 years of age.

**Table 1 ijerph-20-03536-t001:** Descriptive components of the Barcelona Social Health Inequalities Surveillance System.

Indicator	Definition	Sources of Information	Axes of Inequality	Periodicity	Years Analysed	Measures
Social determinants of health						
Income index	Household Disposable Income per Capita (combination of five weighted variables related to a city average centred at 100)	Barcelona City Council	Neighbourhood	Annual	2017	Index compared to city mean centred at 100
Health related with behaviours (diet and physical activity)						
Overweight and obesity	Body Mass Index > 25 (based on declared weight and size) in population ≥ 15 years of age	Catalan Health Survey (sample of Barcelona city)	Sex, age and sex, country of origin and sex, social class and sex	Annual (data of two years grouped)	2011–2020	Prevalence for the last year, RII and SII (or PR and PD for variables with two categories or not ordinal) for trends
Use of healthcare						
Having visited the dentist	Having visited the dentist during the last year in population ≥ 15 years of age	Catalan Health Survey (sample of Barcelona city)	Sex, age and sex, country of origin and sex, social class and sex	Annual (data of two years grouped)	2019–2020	Prevalence
Health outcomes						
Perceived health status	Perception of fair or poor health in population ≥ 15 years of age	Catalan Health Survey (sample of Barcelona city)	Sex, age and sex, country of origin and sex, social class and sex	Annual (data of two years grouped)	2011–2020	Prevalence for the last year, RII and SII (or PR and PD for variables with two categories or not ordinal) for trends
Perceived mental health status	Perception of poor mental health: General Health Questionnaire ≥ 3 [11] in population ≥ 15 years of age	Barcelona Health Survey	Sex, age and sex, country of origin and sex, social class and sex	Quinquennial (every 5 years)	2001–2021	Prevalence for the last year, RII and SII (or PR and PD for variables with two categories or not ordinal) for trends
Life expectancy at birth	Life expectancy at birth smoothed with Bayesian models following Perez-Panades et al. [12]	Mortality registry	Sex, neighbourhood and sex	Annual	1998–2019	Neighbourhoods grouped by quintile of income
Teenage pregnancy	Rates of pregnancy per 1000 women of 15–19 years of age.	Registry of births and abortions	Country of origin, district	Annual	2008–2020	Districts grouped by quintile of income
COVID-19 incidence	Cumulative incidence per 100,000 inhabitants (cases diagnosed by laboratory tests)	COVID-19 registry	Sex, neighbourhood and sex	Annual	2020–2021	Neighbourhoods grouped by quintile of income

RII: relative index of inequality, SII: slope index of inequality, PR: prevalence ratio, PD: prevalence difference.

## Data Availability

Data are presented through the website: https://www.aspb.cat/documents/vigilanciadesigualtats/.

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
