# Peer review of "A City Surveillance System for Social Health Inequalities: The Case of Barcelona"

_ijerph, 2023, doi:10.3390/ijerph20043536_

Round 1
Reviewer 1 Report (Previous Reviewer 5)
It's revised properly. No more comment.
Author Response
Thank you
Reviewer 2 Report (Previous Reviewer 3)
Dear Authors, thank you for your answers for the review.
Author Response
Thank you. English language has been reviewed again.
Reviewer 3 Report (Previous Reviewer 2)
Dear Author,
your article is improved, but still needs some minor adjustments if you and Editors find it relevant.
Kind regards

Author Response
This article is improved yet it needs additional improvements (technical and language). It misses some relevant references for example: “In the city of Barcelona (Spain), public health technicians have promoted the existence of a wealth of research showing health inequalities by gender, social class, geo-graphical area, and other axes of inequality.“
ANSWER TO REVIEWERS’S COMMENT: Thank you, the article has been corrected again by an English person. The references have been added.
Reviewer 4 Report (Previous Reviewer 1)
Dear authors,
Your paper has gained clarity. Nevertheless, some aspects - mentioned below - should be considered.
Line 30-38: First of all, it should be clarified why the term health inequalities is used instead of health inequities. The new introductory section should explain the role of health inequalities monitoring strategies for public health (policy) and health systems and the benefits and harms associated with them.
The section on methods could be still improved. The choice of indicators should be better justified, especially in terms of theoretical or empirical arguments (see e.g. international reports). It would also be important to know what other key indicators would have been available and why they were not selected (especially in relation to the health system, e.g. use of preventive services).
Line 153: The abbreviation CSI-CSV should be explained.
Line 156-166: It should be considered whether this paragraph should not be better placed in the concluding remarks. In addition, the information is also given in lines 202-203.
Line 163: How will the evaluation of whether the system works be made?
Line 183: The arguments are understandable, but the question arises whether a monitoring system should not include an indicator to evaluate the "regular" health system covered by the Spanish national health system.
Line 205-207: Please add specific recommendations on how other governments or stakeholders could proceed with health surveillance based on your profound experience.
Author Response
ANSWER TO REVIEWERS’S COMMENT: Thank you for the comments. Main changes are shown in yellow colour (other changes are due to the English correction)
Line 30-38: First of all, it should be clarified why the term health inequalities is used instead of health inequities. The new introductory section should explain the role of health inequalities monitoring strategies for public health (policy) and health systems and the benefits and harms associated with them.
ANSWER TO REVIEWERS’S COMMENT: Many terms have been used as “disparities”, “inequities”, “inequalities”. We have used the term “inequalities” as it is used in many similar reports, see for example references 3,9,13. In the Introduction now there is a new text explaining a little bit more on the monitoring strategies.
The section on methods could be still improved. The choice of indicators should be better justified, especially in terms of theoretical or empirical arguments (see e.g. international reports). It would also be important to know what other key indicators would have been available and why they were not selected (especially in relation to the health system, e.g. use of preventive services).
ANSWER TO REVIEWERS’S COMMENT: Now we have added a sentence referred to the choice of indicators. In the Concluding remarks we discuss on the aspects related to other indicators referred to the National Health System.
Line 153: The abbreviation CSI-CSV should be explained.
ANSWER TO REVIEWERS’S COMMENT: This is explained at the bottom of the Figure.
Line 156-166: It should be considered whether this paragraph should not be better placed in the concluding remarks. In addition, the information is also given in lines 202-203.
ANSWER TO REVIEWERS’S COMMENT: We are not able to see which paragraph the reviewer refers. But we have revised again the text and we consider that is placed correctly.
Line 163: How will the evaluation of whether the system works be made?
ANSWER TO REVIEWERS’S COMMENT: We have added a sentence in this section.
Line 183: The arguments are understandable, but the question arises whether a monitoring system should not include an indicator to evaluate the "regular" health system covered by the Spanish national health system.
ANSWER TO REVIEWERS’S COMMENT: In the discussion section we explain that if we detect health services inequalities in other health information systems, we can include a new indicator in the system.
Line 205-207: Please add specific recommendations on how other governments or stakeholders could proceed with health surveillance based on your profound experience.
ANSWER TO REVIEWERS’S COMMENT: The recommendation has been included.
This manuscript is a resubmission of an earlier submission. The following is a list of the peer review reports and author responses from that submission.
Round 1
Reviewer 1 Report
Dear authors,
The article is interesting and a good example of how monitoring of health inequalities can be regularly carried out in order to use these data as a basis for decision-making. The methods section should be more specific. For example, a rationale for odontologist as an indicator of health care is only found in the discussion. In this context, it would also have been interesting to see what other indicators would have been available.
Furthermore, a reference to the quality of the data and the handling of the different references of the data (e.g. date of assessment) should be included.
Furthermore, it should be mentioned why the decision was made to use a reduced set of data for monitoring and what restrictions arise for the use of these results. There is only one reference to this in the discussion.
The discussion should be revised and focus much more on content-related arguments. For example, it would be interesting to know what the benefits of monitoring are; what opportunities and limitations result from it.
Author Response
Reviewer 1
The article is interesting and a good example of how monitoring of health inequalities can be regularly carried out in order to use these data as a basis for decision-making. The methods section should be more specific. For example, a rationale for odontologist as an indicator of health care is only found in the discussion. In this context, it would also have been interesting to see what other indicators would have been available.
ANSWER: As we stated, we were interested in having a small number of indicators (as for example has been done in the Marmot review). Following the reviewer’s suggestion, we have explained more deeply the indicators chosen. We feel that is more appropriate to explain that in the section now named “The surveillance system”, since is in this section where we explain how all the steps of the Monitoring System were achieved.
Furthermore, a reference to the quality of the data and the handling of the different references of the data (e.g. date of assessment) should be included.
ANSWER: As it is stated in Table 1 the indicators are from annually data. The only exception is the Barcelona Health Survey that has data for every 5 years. This information now is in a new column of Table 1.
Referred to the quality of data we have now stated in the SOURCES OF INFORMATION section that the surveys are approved by the Statistical Institute of Catalonia, that assures the quality excellence of its products. Moreover, we have explained more deeply other aspects related to the data sources.
Furthermore, it should be mentioned why the decision was made to use a reduced set of data for monitoring and what restrictions arise for the use of these results. There is only one reference to this in the discussion.
ANSWER: According to the reviewer’s comment, we have explained our reasons a little bit more and included other references in the second paragraph of the Discussion.
The discussion should be revised and focus much more on content-related arguments. For example, it would be interesting to know what the benefits of monitoring are; what opportunities and limitations result from it.
ANSWER: We have improved it (penultimate paragraph of the discussion).
Reviewer 2 Report
This article is declared as a short report on the Surveillane system for social health inequalities in the city of Barcelona. The report is very interesting for municipalities and cities that could apply a similar type of monitoring to take appropriate action, but with this approach, in my humble opinion, it is not even possible to plan such an approach.

Author Response
Answer to the reviewer’s comments
This article is declared as a short report on the Surveillance system for social health inequalities in the city of Barcelona. The report is very interesting for municipalities and cities that could apply a similar type of monitoring to take appropriate action, but with this approach, in my humble opinion, it is not even possible to plan such an approach
ANSWER: General comment: It is important to take into account that this article is not an ORIGINAL SCIENTIFIC ARTICLE. The objective of the article is to describe a tool developed for Barcelona city. For this reason, we changed the titles of the sections in order to not include the sections of ORIGINAL ARTICLES. After the introduction there is a section explaining how the System was created (JAHEE JointAction, steps and group of experts) and in the next section the Monitoring System is described. The article is similar to a CASE REPORT to show a specific Public Health Monitoring System.
In the introduction, the authors do not sufficiently address the problem of inequalities in the city of Barcelona and the need to implement this system
ANSWER: We have explained a little bit more that until 2015 all the aspects related with health inequalities in Barcelona were mainly research-based with no political commitment until 2015.
In the Methods section, the authors say nothing about the methodology developed under the Joint Action on Health Equity in Europe (JA - 16 HEE).
ANSWER: As we said above there is not a section of Methods. We have explained a little bit more the JAHEE Joint Action in the section “Establishment of the Surveillance System”.
How was the entire target population of the city of Barcelona covered by this system? Also the children? (This is partly explained in a later section of results, but more details are needed).
ANSWER: As we explain when we describe the system all the population is included. Now we have explained a little bit more. Take into account that some indicators include all population (e.g: life expectancy at birth, COVID incidence). Now we have included in Table 1 the ages analized if they do not include all population.
The whole article is confusing - under the results section you will find the target population, the data analysis and the implementation of the system (under the method section)
Also, the results section describes the areas identified but not how they were selected (methods). In the results, the authors state that the target population is all residents of Barcelona, but the measurements were made through health surveys - this was not described in the methods section. The results are not clearly presented
ANSWER: Please see our general comment above to understand how the manuscript is written. As it is explained in the section “Establishment of the Surveillance System” the experts decided all the aspects by consensus and having reviewed other Systems of Europe (based mainly on the work done in the JAHEE Joint Action).
In the discussion section there are no relevant comparisons of the problem with the literature. The discussion mainly refers to the results section.
ANSWER: This section is aimed to explain the advantages of the System and its future. Now we have named this section as “Concluding remarks” in order not to misunderstand it.
MeSH words are missing.
ANSWER: the key words have been revised and now they are MESH terms.
Other comments of the reviewer:
ANSWER: Please, see our general comment above to understand how the manuscript is organised.
Reviewer 3 Report
Indroductory part of the paper should be enriched (more up to date literature on the theory of health inequalities is required). It seems that healthy life years at birth instead of life expectancy at birth would be more suitable indicator to measure the health inequalities. It might be worth considering using the excess death indicator associated with COVID-19 instead of COVID-19 incidence rate among the city population. As for the axes of health inequalities one should consider adding the level of education of the population in the city. Article needs technical and language editing.
Author Response
Indroductory part of the paper should be enriched (more up to date literature on the theory of health inequalities is required).
ANSWER: As a short article we did not try to deep on the literature of health inequalities. However, following the reviewer’s suggestion, we have added more information in the paragraph 1 of the article.
It seems that healthy life years at birth instead of life expectancy at birth would be more suitable indicator to measure the health inequalities.
ANSWER: The experts analysed this indicator but concluded that to include it was not easy because the health data were not always available for small areas of the city, therefore we decided to include life expectancy as it is a very good indicator and easier to calculate. Moreover, life expectancy at birth is more comparable across different cities because it does not depend on the “health” indicator chosen to calculate Healthy life expectancy.
It might be excess death indicator associated with COVID-19 instead of COVID-19 incidence rate among the city population.
ANSWER: We see the point of the reviewer, but as stated in the paper, COVID-19 incidence has been an important indicator of health inequalities in the city of Barcelona. Anyway, in the near future, if necessary, we are able to change it as stated in the discussion section.
As for the axes of health inequalities one should consider adding the level of education of the population in the city.
ANSWER: The group of experts decided to use occupational social class as an indicator of socioeconomic position. This indicator is proposed by the Spanish Society of Epidemiology and also of Public Health. The level of education does not capture health inequalities in the young population because everybody has reached the secondary education (as it is mandatory).
Article needs technical and language editing.
ANSWER: It has been revised again
Reviewer 4 Report
For the introduction: have other areas created surveillance systems? How do these differ from the one created for Barcelona? How will the surveillance system be used
Methods: This section doesn't tell you how the system was created. For example how was consensus between different professional reached? Was a Delphi method used? All it currently tells you is who was involved in the process. After further reading, in the results, the authors mention use of Poisson regression. So were measures to use in this analysis determined by health professionals
In Table 1: Overweight and obesity is a health outcome not health behaviour
For use of health care, I think you mean dentists not odontologist-these are people who do forensic dentistry after a crime-so an average person wouldn't visit them.
The scope of variables included in the analysis is very narrow. It is possible that by using this for decision making it could increase inequalities. For example for BMI it does not look at differences across the life course which could be important for decision making. What about other health service usage.
In Table 1, it also doesn't state when the data used for the model comes from. Is it all over the same time period? How do the researchers match individuals between data sets. How might their matching strategy impact on the findings?
So is the aim of this research, that practitioners will look at these results to inform decisions? Is annual updating frequent enough?
Author Response
For the introduction: have other areas created surveillance systems? How do these differ from the one created for Barcelona? How will the surveillance system be used
ANSWER: We have included a small text related with the reality of other surveillance systems (first paragraph of the Introduction). Also the penultimate paragraph of the discussion explains the uses of the system.
Methods: This section doesn't tell you how the system was created. For example how was consensus between different professional reached? Was a Delphi method used? All it currently tells you is who was involved in the process.
ANSWER: We have added a text at the end of the section now named “Establishment of the Surveillance System”.
After further reading, in the results, the authors mention use of Poisson regression. So were measures to use in this analysis determined by health professionals
ANSWER: As it is stated in the article, the group of experts included epidemiologists and statisticians and they shared they expertise with the other members of the groups in order to decide the analysis to be used. It is necessary to mention that robust Poisson regression has been proved to be a good approach to estimate prevalence ratios.
In Table 1: Overweight and obesity is a health outcome not health behaviour
ANSWER: We agree with the reviewer and we have changed the text and the label by “Health related with section now named “sebehaviours (diet and physical activity)”
For use of health care, I think you mean dentists not odontologist-these are people who do forensic dentistry after a crime-so an average person wouldn't visit them.
ANSWER: The reviewer is right, we have included the word “dentist” accordingly
The scope of variables included in the analysis is very narrow. It is possible that by using this for decision making it could increase inequalities. For example for BMI it does not look at differences across the life course which could be important for decision making. What about other health service usage.
ANSWER: The system includes information by age-groups and this can be useful to interpret the life-course. As we have stated in the article, we decided to have a small number of indicators because they are presented by several axes of inequalities, therefore there is a lot of information in the system. However, in the future we can change it if we see that the system is not useful enough.
In Table 1, it also doesn't state when the data used for the model comes from. Is it all over the same time period? How do the researchers match individuals between data sets. How might their matching strategy impact on the findings?
ANSWER: We have included in table 1 a column with the years studied until now. We do not match datasets, they are different sets of data.
So is the aim of this research, that practitioners will look at these results to inform decisions? Is annual updating frequent enough?
ANSWER: Now we do not have the possibility to have real time data but we will try to have this possibility for the future (as it is stated in the discussion section). Anyway we believe that annual data for most of the outcomes is enough to inform decisions and plan interventions. We hope that this system will be used by different stakeholders as it is stated in the “Dissemination” paragraph.
Reviewer 5 Report
Establishing a city surveillance system for social health inequalities is really a very important system. This manuscript clearly described the system, and the data collection and analysis for health inequality evaluation. It is really an interesting manuscript. Not only the use of data and showing the health inequalities, but also this would be an important reference for other countries and districts. In this case, there are some suggestions to authors.
1. Regarding the structure of this manuscript, it is mainly to introduce the surveillance system. not analyze the situation of health inequality in Barcelona. So, I suggest the section of method could be changed as establishment of the system.
2. In the surveillance system, there are two important data sources, Barcelona Health Survey and Catalan Health Survey. They need to be more detailed introduced, especially the methods and sample size of population sampleing. Does these two surveys cover all districts and neighbourhoods? If not, how to evaluation the inequalities among districts and neighbourhoods?
3. The paragrah about surveilance indicators development, indices sellected for inequalities, etc. (now in the section of results) should be moved to section of surveillance system establishment.
Author Response
Reviewer 5
Establishing a city surveillance system for social health inequalities is really a very important system. This manuscript clearly described the system, and the data collection and analysis for health inequality evaluation. It is really an interesting manuscript. Not only the use of data and showing the health inequalities, but also this would be an important reference for other countries and districts. In this case, there are some suggestions to authors.
- Regarding the structure of this manuscript, it is mainly to introduce the surveillance system. not analyze the situation of health inequality in Barcelona. So, I suggest the section of method could be changed as establishment of the system.
ANSWER: We agree that the sections of original manuscripts are not good for this article that shows an example of a Surveillance System. For this reason, following the reviewer’s suggestion, we have changed the title of the sections in order to be more clear
- In the surveillance system, there are two important data sources, Barcelona Health Survey and Catalan Health Survey. They need to be more detailed introduced, especially the methods and sample size of population sampling. Do these two surveys cover all districts and neighbourhoods? If not, how to evaluation the inequalities among districts and neighbourhoods?
ANSWER: We have added detailed information regarding the two data sources in the SOURCES OF INFORMATION section.
- The paragraph about surveillance indicators development, indices selected for inequalities, etc. (now in the section of results) should be moved to section of surveillance system establishment.
ANSWER: We think that is better to include all the steps related with the system in the same section, but now it has been named: “The Surveillance System”.